ecology

anthrax, disease dynamics, ecological trap, environmental transmission, habitat selection, transmission hotspot

**Author for correspondence:**
Yen-Hua Huang
e-mail: yhhuang0324@gmail.com

# Disease or drought: environmental fluctuations release zebra from a potential pathogen-triggered ecological trap

Yen-Hua Huang[1,10], Hendrina Joel[2], Martina Küsters[3], Zoe R. Barandongo[1,10], Claudine C. Cloete[4], Axel Hartmann[4], Pauline L. Kamath[5], J. Werner Kilian[4], John K. E. Mfune[2], Gabriel Shatumbu[4], Royi Zidon[6], Wayne M. Getz[7,8] and Wendy C. Turner[9,10]

[1]Department of Forest and Wildlife Ecology, University of Wisconsin-Madison, Madison, WI 53706, USA
[2]Department of Biological Sciences, University of Namibia, Windhoek, Namibia
[3]Okaukuejo, Namibia
[4]Etosha Ecological Institute, Ministry of Environment, Forestry and Tourism, Okaukuejo, Namibia
[5]School of Food and Agriculture, University of Maine, Orono, ME 04469, USA
[6]Institute of Earth Sciences, Hebrew University of Jerusalem, Jerusalem, Israel
[7]Department of Environmental Science, Policy and Management, University of California, Berkeley, CA 94704, USA
[8]School of Mathematical Sciences, University of KwaZulu-Natal, Durban, South Africa
[9]US Geological Survey, Wisconsin Cooperative Wildlife Research Unit, Department of Forest and Wildlife Ecology, University of Wisconsin-Madison, Madison, WI 53706, USA
[10]Department of Biological Sciences, University at Albany, State University of New York, Albany, NY 12222, USA

Y-HH, 0000-0002-2961-0895; ZRB, 0000-0001-6332-991X; PLK, 0000-0002-6458-4514; WMG, 0000-0001-8784-9354; WCT, 0000-0002-0302-1646

When a transmission hotspot for an environmentally persistent pathogen establishes in otherwise high-quality habitat, the disease may exert a strong impact on a host population. However, fluctuating environmental conditions lead to heterogeneity in habitat quality and animal habitat preference, which may interrupt the overlap between selected and risky habitats. We evaluated spatio-temporal patterns in anthrax mortalities in a plains zebra (*Equus quagga*) population in Etosha National Park, Namibia, incorporating remote-sensing and host telemetry data. A higher proportion of anthrax mortalities of herbivores was detected in open habitats than in other habitat types. Resource selection functions showed that the zebra population shifted habitat selection in response to changes in rainfall and vegetation productivity. Average to high rainfall years supported larger anthrax outbreaks, with animals congregating in preferred open habitats, while a severe drought forced animals into otherwise less preferred habitats, leading to few anthrax mortalities. Thus, the timing of anthrax outbreaks was congruent with preference for open plains habitats and a corresponding increase in pathogen exposure. Given shifts in habitat preference, the overlap in high-quality habitat and high-risk habitat is intermittent, reducing the adverse consequences for the population.

## 1. Introduction

Habitat quality is context-dependent [1], where consumers distribute in response to resource dynamics on a landscape [2–5]. This habitat heterogeneity in turn affects disease risk owing to uneven distributions in species diversity of hosts, vectors and reservoirs [6–8], parasite loads [8,9] and abiotic variation [7,8]. Moving across diverse and dynamic landscapes, mobile hosts may encounter different rates of pathogen exposure across habitats [10,11]. Bridging disease dynamics and animal habitat use, therefore, may help understand variation in transmission dynamics [12].

Reservoir dynamics of environmentally transmitted pathogens are likely to depend upon spatial structure, which would in turn be affected by habitat heterogeneity. Aggregation of animal hosts in preferred habitats may 'seed' the environment with a higher density of pathogen propagules, maintaining and spreading infection within higher resource quality habitats [13]. Because pathogens are unlikely to be detectable by hosts, transmission hotspots could establish in otherwise high-quality habitat if animals are attracted to habitat with heightened disease risk [14]. This habitat would be considered an 'ecological trap' if habitat selection preferences lead to lower fitness [15–17]. Ecological trap theory has mainly focused on the effects of predation, food scarcity or anthropogenic disturbance on off-spring survival [16–18]. However, habitats with high concentrations of environmentally transmitted pathogens might also have potential to form ecological traps [13,14].

This study aims to understand the effects of habitat quality and disease exposure on a susceptible and mobile host population, coupling habitat heterogeneity in disease risk, dynamics in habitat quality and host habitat use. We evaluate whether a population of plains zebra (*Equus quagga*) susceptible to anthrax experiences an overlap in high-quality habitat and transmission hotspots, which could potentially lead to the formation of an ecological trap. Anthrax is a highly lethal, acute to peracute disease caused by the bacterium, *Bacillus anthracis*. Given its lethality, anthrax has potential fitness consequences for the zebra population, by reducing average fitness of individuals. Although an earlier demographic study concluded that anthrax may be limiting growth of this zebra population [19], we cannot assess whether an anthrax transmission hotspot is also an ecological trap, because we did not measure fitness differences among habitats.

Anthrax only transmits via the environment, and not directly between hosts [20], and hence provides an opportunity to evaluate the spatial variation in pathogen exposure. Transmission relies upon contact with long-lived spores in soil environmental reservoirs. These reservoirs are generated from host disease mortality and subsequent exposure of susceptible hosts through foraging at *B. anthracis*-laden carcass sites [21–24]. Thus, anthrax is a good candidate for investigating how an infectious disease could turn preferred habitat into a disease transmission hotspot.

Here, we explore the overlap between host habitat selection and areas of heightened anthrax risk, and how preference and risk change with variation in resource availability. We hypothesized that (i) habitat quality is variable in time (i.e. owing to amount of annual precipitation), (ii) the ranking of available habitats by quality and attractiveness is not static, but varies based on context, and (iii) for an environmentally persistent and highly lethal pathogen with only environmental transmission, animals are subject to an overlap in habitats of both high quality and high disease risk—but only sometimes, depending on environmental conditions. Periods of environmental fluctuation that allow reprieve from the transmission hotspot will reduce exposure and the number of pathogen-induced mortalities affecting the population. We first assessed variation in risk of contacting *B. anthracis* by habitat type. We then tested the relationships among habitat dynamics, zebra resource selection among habitats with differential risk, and anthrax mortalities, to evaluate how fluctuating habitat quality affects host habitat use and pathogen exposure, and in turn disease dynamics.

# 2 Methods

## (a) Study area and periods

Our study was conducted during two time periods (2009–2010; 2018–2020) in Etosha National Park (ENP), a fenced 22 270 km² reserve located in northern Namibia (figure 1a). ENP is a semi-arid savannah, with three seasons: cool dry season in May–August, hot semi-dry season in September–December and hot wet season in January–April. Rainfall is strongly seasonal and occurs mainly between November and April, with the greatest monthly rainfall occurring in January and February [25]. There is a west–east rainfall gradient, increasing from an average of 200 to 450 mm yr⁻¹. The average annual rainfall in the central area is 358.0 ± 126.7 mm (mean ± s.d.; Okaukuejo station 1954–2020). Rainfall is recorded by rainfall years (e.g. July 2009–June 2010 is the rainfall year 2010), not calendar years. The specific dates of study were May 2009–August 2010 and October 2018–April 2020, during which we collected telemetry data from collared zebras. Precipitation in 2010 and 2020 was around or above the average (389.9 and 440.5 mm at Okaukuejo station, respectively), but 2019 was by far the driest year on record in ENP (83.7 mm; Okaukuejo station; electronic supplementary material, figure S1a). The 2019 drought was the most severe drought in Namibia in the last 90 years [26].

Much of ENP is covered by mopane (*Colophospermum mopane*) shrubveld or treeveld, and large salt pans, with open grasslands along the pans. There are seven basic vegetation types described, including bare ground, grassland, steppe, grass savannah, shrub savannah, low tree savannah and high tree savannah [27,28]. For this study, we grouped these seven categories into four basic habitat types: bare areas (i.e. salt pans), open plains (grassland, grass savannah and steppe), shrublands (shrub savannah) and woodlands (high tree and low tree savannah; figure 1a). These habitat types were used to evaluate resource fluctuation, disease risk and resource selection.

## (b) Anthrax risk by habitat

Anthrax is endemic in ENP where plains zebra is the most common host detected, constituting more than 50% of cases [29]. An estimated cause of death for every observed mortality is assigned by an investigation of the carcass remains by park staff or researchers, and blood swabs for disease diagnosis are collected regardless of the suspected cause of death. Anthrax mortality in this study was defined as laboratory-confirmed cases from bacterial culture and suspected cases based on symptoms if diagnostic samples were not collected. Signs of suspected anthrax mortalities include no evidence of predation, blood exudation and lack of clotting, and rigidly extended fore legs or oedematous swelling [30]. Zebra anthrax mortalities occur annually with strong seasonality in cases, which peak in the late wet season (March–April) [29,31], and case numbers are positively correlated with annual rainfall [29] (electronic supplementary material, figure S1b). Anthrax transmission relies on environmental reservoirs created by positive carcasses [21,22]. Hence, more anthrax carcasses in an area represent a higher risk of exposure.

We used animal mortalities detected by opportunistic surveillance [32] to evaluate anthrax risk by habitat type. Opportunistic mortality surveillance can be biased towards certain landscape features such as access along the road network or vegetation density [33]. Surveillance may also be biased towards detecting certain types of mortality sources based on how long the carcass remains on the landscape. Given the challenges in

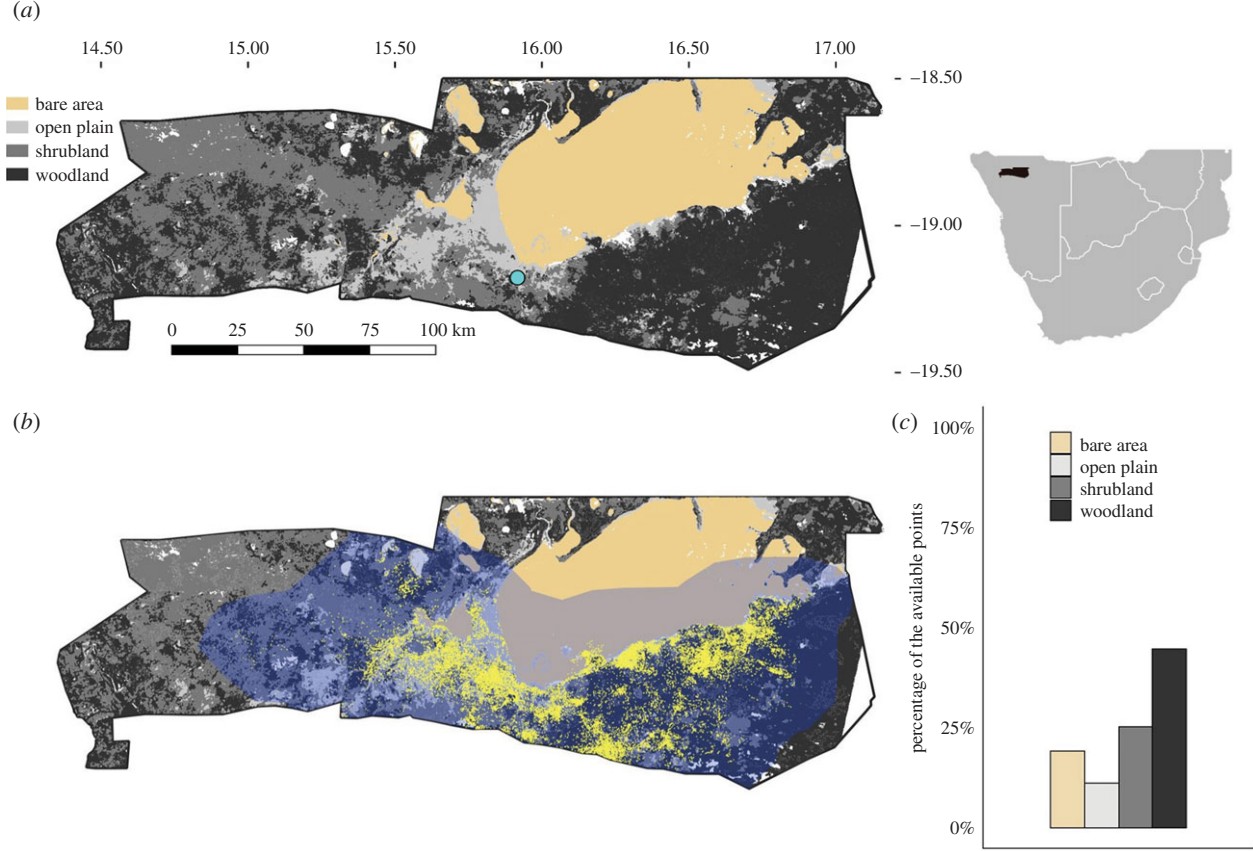

**Figure 1.** The study area of Etosha National Park, Namibia, in southern Africa. (*a*) The distribution of habitat types. The four habitat types considered included open plains (light grey), shrublands (medium grey), woodlands (dark grey) and bare areas (large salt pans; tan). White represents areas with vegetation type not available, which were removed from this study. The blue circle indicates the location of Okaukuejo station. (*b*) Plains zebra (*E. quagga*) space use shows the overall thinned readings of zebra telemetry data (yellow) and their 99.9% kernel density range (blue) which were used to generate available (not used) points for analysis of resource selection functions. (*c*) The percentages of the available points randomly generated within 99.9% kernel density range by habitat type.

detecting carcasses equally across the landscape, we focus on the ratio of anthrax cases to other mortalities, as an index of disease risk among habitats. This method assumes that detection rates of carcasses do not differ among habitat types based on the cause of death. We probed this assumption by comparing a subset of the mortality data during periods of heightened surveillance effort that would reduce detection biases. During the periods represented in this study, there was an additional layer of mortality surveillance based on sites where global positioning system (GPS)-tagged scavengers clustered [34–36]. This additional effort yielded mortality records in locations that otherwise would go undetected, and provided a better estimate of the number of anthrax mortalities, and their temporal and spatial distribution during the study periods. The patterns detected during heightened surveillance effort (electronic supplementary material, figure S2) reflected those in the larger, opportunistic dataset, and thus we feel that the patterns of relative anthrax risk among habitats are robust to differences in surveillance effort.

We compared anthrax mortalities to mortalities from other natural causes (e.g. predation, old age, starvation), to investigate anthrax risk in a habitat type. Using mortalities with recorded GPS positions for herbivore species that had at least one anthrax case from 1998 to April 2020, we summed the numbers of anthrax and other mortalities by habitat for all herbivore species. Bare areas were excluded from these comparisons because animals infrequently used these areas, and no mortality surveillance occurred on the salt pans. In total, there were 737 anthrax mortalities, and 28.5% of them were suspected cases. We used $\chi^2$-tests of independence to examine differences in ratios of anthrax to other mortalities between habitat types, and further examined the differences by herbivore functional

foraging type (grazing, mixed feeding and browsing). For the total mortalities and every foraging type, we conducted three $\chi^2$-tests with two habitat types selected for each, and corrected *p*-values with the Bonferroni correction.

## (c) Dynamics in zebra anthrax cases and habitat quality

We counted accumulated numbers of zebra anthrax mortalities during the seasons involved in the study periods (2009–2010; 2018–2020), when there was enhanced surveillance, to assess spatio-temporal patterns in anthrax mortality in the zebra population. We summed the numbers of anthrax mortalities within each season, and further separated them based on habitat types.

We used a remotely sensed index of vegetation primary production, the fraction of photosynthetically active radiation (FPAR) from Moderate Resolution Imaging Spectroradiometer (Terra MODIS; MOD15A2H) [37] to assess habitat dynamics and variation in quality during the study periods. FPAR is widely used to model vegetation biomass and productivity [27,38]. Although it is subject to the change of foliage resulting in different values across habitat types [27,39], FPAR is a better predictor of grass biomass in ENP than other remotely sensed data sources [27]. The spatial and temporal resolution was 500 × 500 m and 8 days starting at the first day of each year. We extracted FPAR values for the study area (excluding salt pans) and sampling periods, and calculated averages for each habitat type and season.

## (d) Zebra habitat use

Zebra preferentially select shorter, more palatable grasses in open plains [28,40], but in ENP, these grasses get depleted in the dry

season. To determine host habitat selection changes in response to resource fluctuation, we collected telemetry data with GPS collars (African Wildlife Tracking, Pretoria, South Africa) on adult zebras (nine females July 2009–August 2010, eight males and 10 females October 2018–August 2019 and six males and 11 females September 2019–April 2020) captured in central-eastern ENP.

We divided days into morning (6.00–12.00; GMT+1), afternoon (12.00–18.00) and night (18.00–6.00), and thinned the data by extracting readings closest to 9.00, 15.00 and 24.00 for the three periods of a day for each individual, to reduce autocorrelation in the telemetry data. The distance between locations of two consecutive readings of the thinned data could potentially be far enough for an individual to switch habitats (electronic supplementary material, figure S3). If readings for an individual were fewer than 180 in a season (i.e. two months of sampling), the individual's data for that season were removed. Of thinned readings, 6.8% were in areas where vegetation types were not available (figure 1), and these readings were excluded from habitat selection analyses.

We calculated seasonal habitat use for the population, as the percentage of readings in a habitat type by season. We then evaluated habitat selection with resource selection functions (RSFs), based on a use-availability framework [41] to account for the habitat availability in the area. These RSFs correspond to second-order selection, or how individuals select ranges compared to the habitats available in the overall population range [42]. Used versus available (not used) habitat types were then compared with logistic regressions. To define an area with sufficient habitat connectivity from which to generate available locations, we applied a 99.9% kernel density home range estimation to the overall thinned readings from study animals (24 916 readings from 35 individuals) throughout the entire study (clipped by park fenceline; figure 1). The available points were generated with 10 times the thinned readings. Individual-based RSFs were performed to compare the habitats of used and available points [43] in each season, with habitat type as a covariate. We used three orthogonal contrast coding variables for the four-level categorical habitat covariate. The three contrast coding variables compared the preferences for (i) vegetated habitats over bare areas, (ii) open habitats (open plains) over closed habitats (shrublands and woodlands), and (iii) shrublands over woodlands. A regression coefficient represented relative selection strength of an individual [44], which in this case showed habitat preference for a specific habitat comparing to the other habitat type(s). To evaluate whether the RSF findings were robust to the thinning approach selected, we compared these results with RSF results for fixed intervals of 1–24 h. The results of the thinning approach we applied were consistent with fixed intervals of 1–8 h (electronic supplementary material, figure S4).

We fit medians of relative selection strengths by season ($n = 9$ seasons) for the three different comparisons to FPAR with linear regressions, to evaluate the relationships between habitat selection and habitat dynamics. Because FPAR varied with habitat types, we used average FPAR at open habitats as an index of habitat dynamics.

In addition to habitat preference, we investigated how consistently zebras used open habitats, to evaluate how likely an individual would be to die in the open habitats after infected there. We explored the probability that an individual exposed in open habitats would remain in that habitat for the duration of the incubation period (electronic supplementary material, figure S5). From this, we conclude that although daily movement distances can be relatively long (out to 9 km h$^{-1}$; electronic supplementary material, figure S3) [45], the average distances are much smaller (the median across individuals and seasons: 0.26 km h$^{-1}$; electronic supplementary material, figure S3), and zebras have the highest probability of remaining in open habitats

when anthrax cases are most prevalent (75% in wet season; electronic supplementary material, figure S5).

## (e) Integrating dynamics in habitats, host selection and anthrax mortality

We evaluated whether seasonal variation in habitat dynamics or host habitat selection could predict the number of anthrax mortalities. We fit anthrax case numbers by season ($n = 9$ seasons) with linear regression separately to FPAR and the median of relative selection strength. We chose the relative selection strength corresponding to the selection of high- versus low-risk habitats, based on the results of analysis for anthrax risk by habitat. The numbers of anthrax mortalities were square-root transformed owing to overdispersion and small sample size of the dataset. As in the previous analysis, here average FPAR at open habitats was used as the index of habitat dynamics.

All analyses were done in R v. 3.6.1 [46]. FPAR was downloaded from the National Aeronautics and Space Administration (NASA) Land Processes Distributed Active Archive Center by package MODIStsp [47], and values were extracted by the package raster [48]. Kernel density range was estimated using the package adehabitatHR [49], and clipped by fenceline with the package rgeos [50]. Available points were generated with the package sf [51]. Package sp [52,53] was used to retrieve vegetation/habitat types for used and available points. $\chi^2$-tests, Bonferroni correction, logistic regression and linear regression were done using the package stats [46].

## 3. Results

### (a) Anthrax risk by habitat

In general, more herbivore mortalities were found in open habitats than shrublands or woodlands, and anthrax mortality risk was highest in open habitats (figure 2). Mortality data from all herbivorous anthrax host species showed relatively more anthrax mortalities than other mortality sources in open plains than in shrublands ($\chi^2_1 = 14.36$, $p < 0.001$; figure 2a) or woodlands ($\chi^2_1 = 91.54$, $p < 0.001$; figure 2a), and relatively more anthrax mortalities in shrublands than in woodlands ($\chi^2_1 = 32.03$, $p < 0.001$; figure 2a). Comparing host species by functional foraging types, grazing herbivores had a higher proportion of anthrax mortalities in open plains than in woodlands ($\chi^2_1 = 14.93$, $p < 0.001$; figure 2b) and a higher proportion in shrublands than in woodlands ($\chi^2_1 = 10.36$, $p < 0.01$; figure 2b). Mixed-feeding herbivores also had a higher proportion of anthrax mortalities in open plains than in woodlands ($\chi^2_1 = 10.36$, $p < 0.01$; figure 2c). Browsing herbivores rarely died of anthrax, with no significant differences in the proportion of anthrax mortalities between habitats (figure 2d). More browsing herbivore mortalities were detected in closed than in open habitats, in contrast with higher carcass detection in open habitats for other foraging types. These patterns of relative anthrax risk among habitats for the longer time series (1998–April 2020) were corroborated by patterns observed during the shorter periods of increased surveillance effort (electronic supplementary material, figure S2).

### (b) Dynamics in zebra anthrax cases and in habitat quality

Peaks in total zebra anthrax mortalities correlated with seasonal peaks in vegetation productivity in wet seasons (figure 3), with more cases in open habitats than closed habitats at those

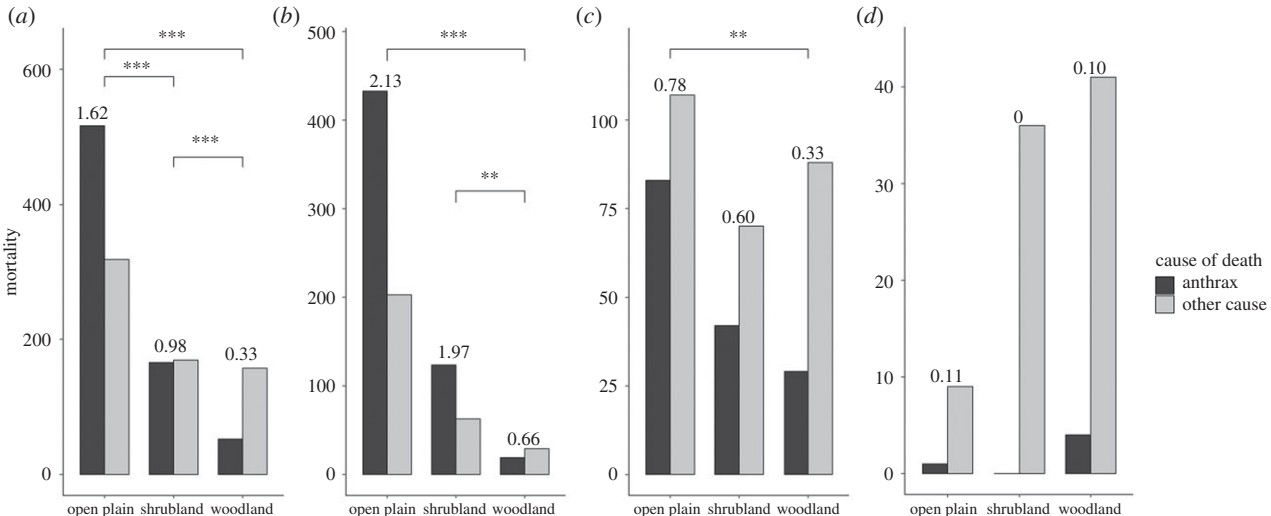

**Figure 2.** Herbivore mortalities by habitat type in Etosha National Park. Mortalities from 1998 to April 2020 with GPS locations were assigned to habitat type (open plains, shrublands or woodlands) and cause of death (anthrax or other natural mortalities) for herbivore species which had at least one anthrax mortality. The mortalities are summed for (*a*) all herbivore species, and then grouped by three functional foraging types, (*b*) grazers, including plains zebra (67.6% of total cases), blue wildebeest (*Connochaetes taurinus*; 9.9%) and gemsbok (*Oryx gazella*; 1.1%), (*c*) mixed feeders including springbok (*Antidorcas marsupialis*; 12.3% of total cases) and African elephant (*Loxodonta africana*; 8.5%), and (*d*) browsers including greater kudu (*Tragelaphus strepsiceros*; 0.3% of total cases) and black rhino (*Diceros bicornis*; 0.4%). The numbers above bars indicate ratios of anthrax mortalities to other natural causes of death. The asterisks show the significance of $\chi^2$-tests comparing the proportions of anthrax mortalities between paired habitat types. Two and three asterisks represent the Bonferroni corrected *p*-values less than 0.01 and 0.001, respectively.

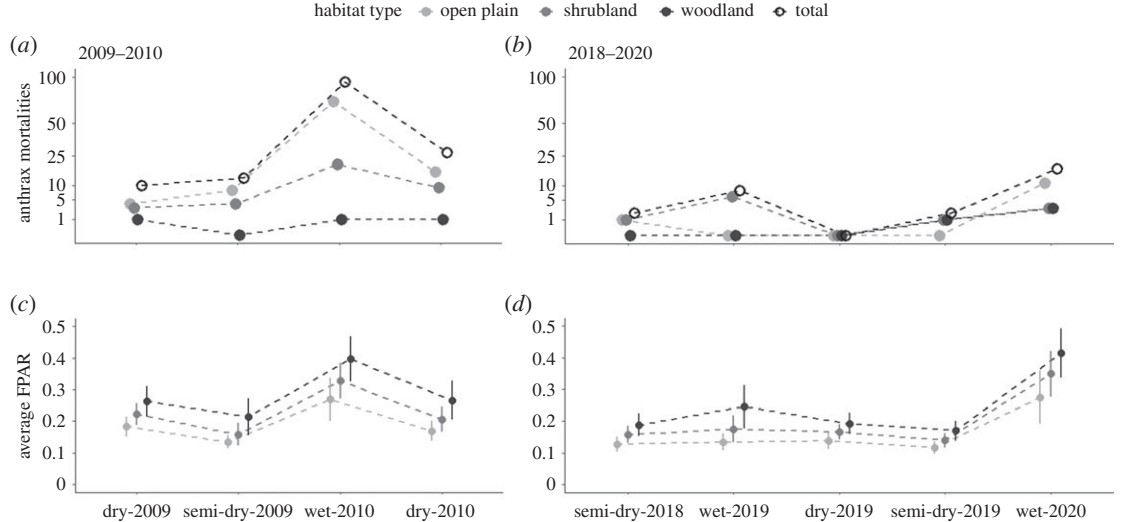

**Figure 3.** Zebra anthrax mortalities (*a,b*) and vegetation productivity (*c,d*) in Etosha National Park, by habitat type and season in 2009–2010 (*a,c*) and 2018–2020 (*b,d*). Productivity was measured as the FPAR. Error bars indicate standard deviations of spatial variation. The axes for anthrax mortalities were square-root transformed. The study period 2009–2010 was an average rainfall year; 2018–2020, a severe drought followed by an above average rainfall year.

productivity peaks. However, inter-annual rainfall patterns affected the intensity of these seasonal anthrax outbreaks. Reduced primary production during the 2019 drought corresponded with very few anthrax deaths: there were only 12 zebra anthrax mortalities recorded in 2018–2019, with eight in the wet season (figure 3). The average and above average rainfall years had larger anthrax outbreaks but at different scales, with 144 zebra anthrax mortalities in 2009–2010 (94 cases in the wet season; figure 3) and 18 zebra anthrax mortalities in the wet season of 2020 (figure 3), respectively.

## (c) Zebra habitat use
Zebra habitat selection preferences varied with rainfall amount and fluctuations in primary production, except for

a consistent avoidance of bare areas ($R^2 = 0.006$, $t = -0.21$, $p = 0.837$, $n = 9$; figures 4 and 5). Zebra predominantly used open plains over any closed habitats in the wet season of years with average and above average rainfall when there was higher vegetation productivity (figure 4*a,b*). However, during the semi-dry season in 2009 and during the 2019 drought, vegetation productivity was reduced, and zebra primarily used woodlands (figure 4*a,b*). In average and above average rainfall years, zebra showed tendencies to select open habitats over closed habitats and shrublands over woodlands, especially in the wet season (figures 4 and 5). By contrast, zebra selected closed habitats and woodlands during the 2019 drought (figures 4 and 5). Both relative selection strengths for open over closed habitats and for

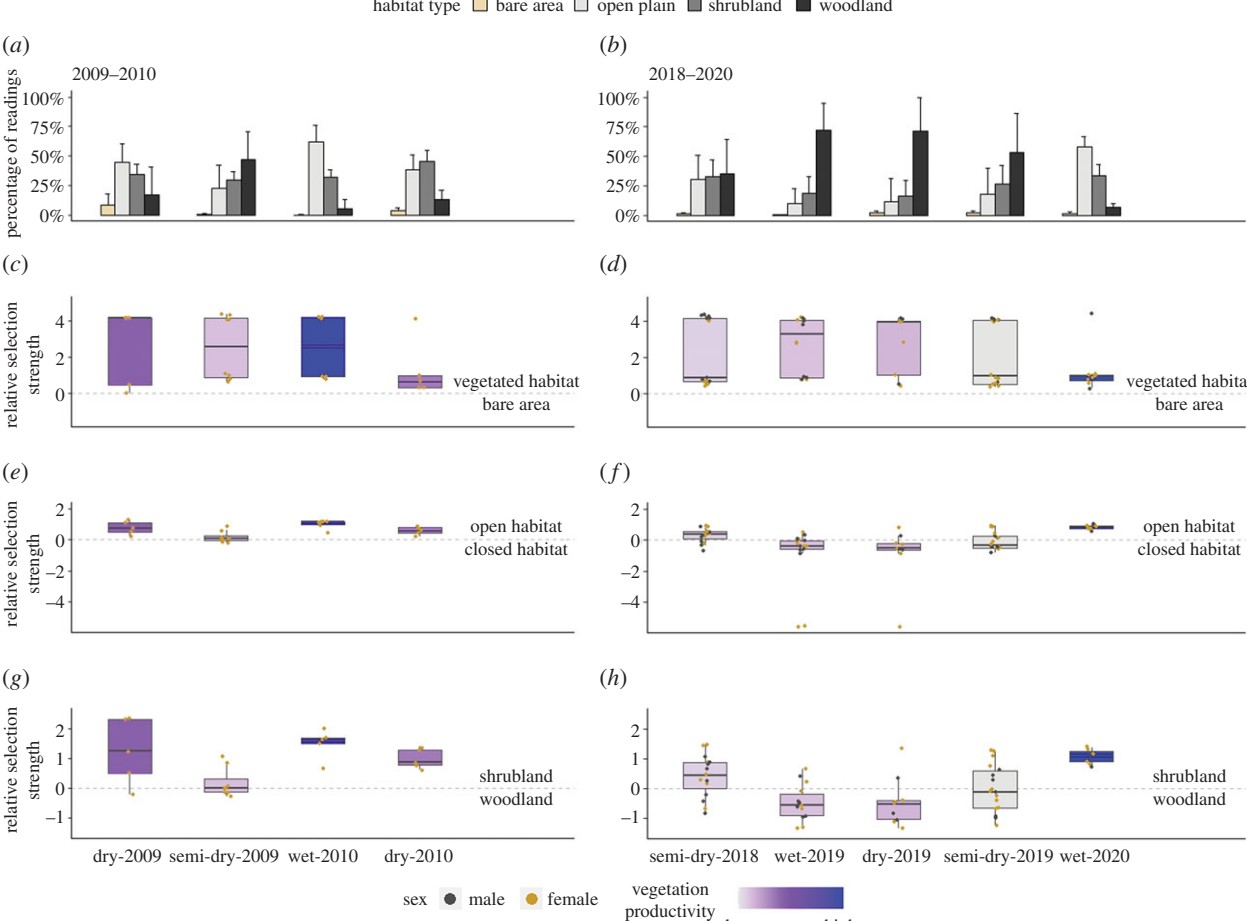

**Figure 4.** Inter-annual and seasonal variation in zebra habitat selection in Etosha National Park. (*a,b*) The average percentages of zebra selected locations among habitat types by season (*a*) in 2009–2010 (an average rainfall year) and (*b*) in 2018–2020 (a drought year, 2019, followed by an above average rainfall year, 2020), with error bars representing standard deviations from individual differences. (*c–h*) The regression coefficients indicating zebra habitat selection accounting for habitat availability by season and year. The coefficients of three contrast coding variables from seasonal individual-based RSFs represented the relative selection strength (habitat preference) for vegetated habitats over bare areas (*c*) in 2009–2010 and (*d*) in 2018–2020, for open habitats over closed habitats (*e*) in 2009–2010 and (*f*) in 2018–2020, and for shrublands over woodlands (*g*) in 2009–2010 and (*h*) in 2018–2020. Each point is an individual zebra, with males in grey and females in yellow. In 2009–2010, all the collared zebras were female. Sample sizes of individual zebras per season ranged from 5 to 17. The boxplots in (*c–h*) are colour-coded with the seasonal average values of FPAR at open habitats reflecting vegetation productivity differences by season, with grey to purple to blue representing low to high FPAR.

shrublands over woodlands were significantly positively related to FPAR ($R^2 = 0.67$, $t = 3.75$, $p < 0.01$, $n = 9$; $R^2 = 0.63$, $t = 3.43$, $p < 0.05$, $n = 9$; figure 5*b,c*). No obvious sex differences in habitat selection were detected (figure 4), though sample sizes were relatively small.

## (d) Integrating dynamics in habitats, host selection and anthrax mortality

The number of anthrax cases recorded in a season can be linked to the amount of primary productivity and the resulting host habitat selection preferences (figure 6). There was a significantly positive relationship between primary production and the square-root transformed number of anthrax cases recorded per season ($R^2 = 0.56$, $t = 2.97$, $p < 0.05$, $n = 9$; figure 6*a*). The relative selection strength shown by zebras for open habitats over closed habitats also associated with the number of anthrax mortalities ($R^2 = 0.60$, $t = 3.26$, $p < 0.05$, $n = 9$; figure 6*b*), where the stronger the preference for open habitats, the more anthrax cases were recorded.

## 4. Discussion

This study related resource dynamics and host habitat preferences to temporal and spatial variation in disease outbreak dynamics. Host habitat selection varied among seasons and rainfall years in response to environmental fluctuations and habitat dynamics. Zebra preferred open habitats with higher anthrax risk in wet seasons and wetter years, and showed correspondingly higher anthrax mortality associated with higher primary production. In dry seasons and drought, zebras shifted their selection preference away from risky habitats, with a corresponding reduction in anthrax mortality. Our results suggest that habitat dynamics and host habitat selection can be used to predict disease outbreaks for environmentally transmitted disease agents.

The associations detected between habitat selection and disease risk are supported by seasonal differences in zebra diet selection [40] and pathogen exposure [54]. Together, these indicate that a disease transmission hotspot has developed in the open habitat preferred by zebras. Given the long-lived nature of *B. anthracis* spores, pathogen reservoirs can survive periods of low host density, and infect zebras

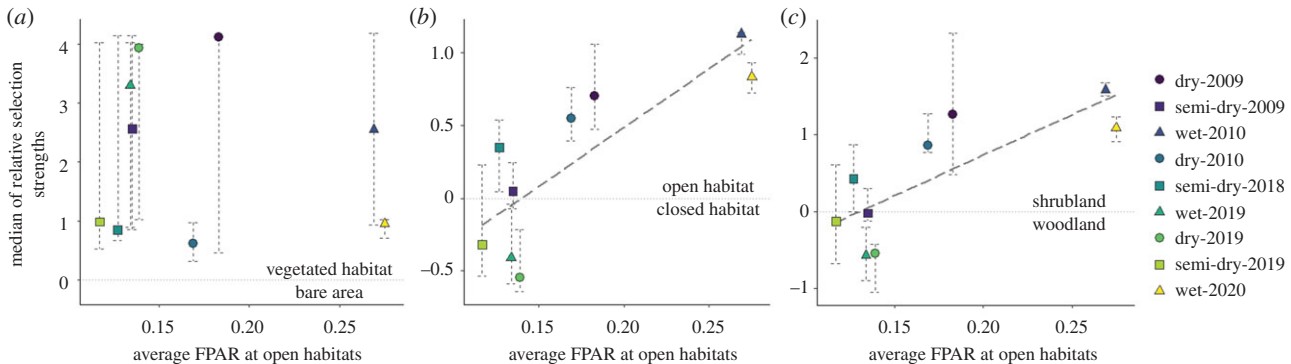

**Figure 5.** Seasonal relationships between habitat dynamics and zebra habitat selection. Medians of relative habitat selection strengths in relation to average FPAR (a remote-sensing index of vegetation productivity) at open habitats (*a*) for vegetated habitats over bare areas, (*b*) for open habitats over closed habitats and (*c*) for shrublands over woodlands. An error bar indicates interquartile range from individual variation in each season. Grey dashed lines are best-fitting lines when linear regressions showed significant slopes. Different years/seasons are colour-/shape-coded.

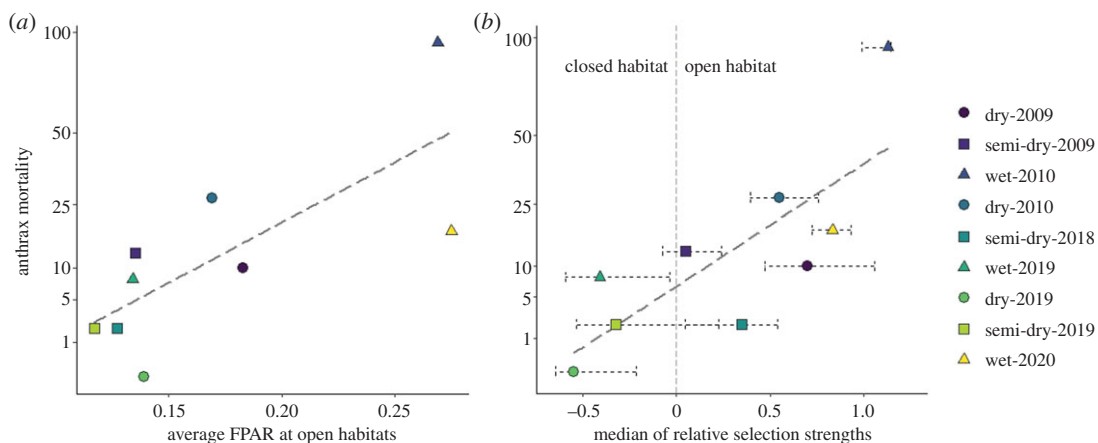

**Figure 6.** Linear regressions between habitat dynamics, zebra habitat selection and zebra anthrax cases. Relationships are shown between the square-root transformed anthrax case numbers by season in relation to (*a*) average FPAR (a remote-sensing index of vegetation productivity) at open habitats, and (*b*) the median of relative selection strengths for open habitats over closed habitats. Error bars indicate interquartile ranges of relative selection strengths from individual variation. Grey dashed lines are best-fitting lines from linear regressions. Different years/seasons are colour-/shape-coded.

when conditions support their return to this habitat. The short latency of infection and highly lethal nature of this disease [20,55] is likely to maintain the transmission hotspot through a positive feedback loop. Zebras contracting the disease on the plains will probably die in these areas and create more local infectious zones, especially in the anthrax season (electronic supplementary material, figure S5), and hence enhance pathogen concentrations in the hotspot habitats. The lack of fitness data prevented us from determining whether this transmission hotspot is also an ecological trap. However, because the transmission hotspot and preferred habitat only overlapped when resources were abundant, even if an ecological trap were to form, it would be temporary, and the detrimental effect on the host population reduced owing to the reprieve of selecting different habitats in different periods.

Grazing herbivores may travel long distances, seeking areas with forage availability in response to a drought when grass biomass in preferred locations is depleted [56]. The zebra population selected the open plains in the wet season in the average to above average rainfall years, and it switched to woodlands in the drought. The plains have palatable grasses which zebra preferentially select [40], but they get depleted in the dry season most years, forcing individuals to use other habitats [45]. During a drought, grass biomass in the open plains does not recover, which prevents its selection by zebra. As a result, they occupy closed habitats where otherwise less palatable vegetation is found [28,40]. Consequently, zebra use high disease risk habitats when their resources are abundant, and lower risk habitats when the more desirable habitat is depleted.

The spatial structure of exposure risk and host habitat selection is most likely to determine outbreak intensity in the zebra population rather than other potential factors, such as stress owing to nutritional deprivation. These results imply that the zebra population never gets a really 'excellent' year because it suffers either from a deadly disease or from food scarcity. However, the population also probably benefits from this shifting habitat selection, which staggers two negative impacts. The population is exposed to higher risk areas when resources are abundant and individuals are less stressed than in the dry periods [57]. This suggests that better health as a result of more resource availability may play a factor in preventing massive anthrax outbreaks in locations like ENP with wet-season outbreaks. In dry-season anthrax outbreaks, which can be more common in other locations, hundreds of animals per species can be impacted [58–60], and species-specific mortality rates in these outbreaks can reach higher than 90% [59].

In comparison, in the wet-season outbreak in 2010 (the biggest outbreak in ENP in nearly 40 years), the zebra anthrax mortality rate is estimated to be around 3.9%. This calculation is based upon observed cases corrected with an estimated total to an observed ratio of 3.8 (2.9–8.2) [32] and a population estimate of 9225 (5138–13 672) in the 99.9% kernel density range (figure 1; Etosha Ecological Institute unpublished aerial survey data from 2005).

The anthrax outbreak in 2010 was the largest in ENP since the early 1970s, and spilled over to the dry seasons. There was also an outbreak in the wet season of 2020, though at a smaller scale. The variation in outbreak sizes between 2010 and 2020 may be attributed to factors such as multi-year rainfall patterns. Though spores can persist in the environment for decades, spore concentrations at a site decay over time, and carcass sites are most infectious within the first few years after host death [21]. Zebra anthrax cases are positively correlated with annual rainfall [29]. Hence, periods of above average rainfall and higher zebra anthrax mortality will increase the number of highly infectious reservoir sites, while dry periods will deplete them. High rainfall years with more zebra anthrax cases preceded the large outbreak recorded in 2010, while drier years with fewer anthrax cases preceded the smaller outbreak in 2020 (electronic supplementary material, figure S1a).

This study provides new insight into spatio-temporal disease dynamics in mobile hosts. The congruence of anthrax outbreaks and host habitat preferences in this study suggests that when disease risk on a landscape is heterogeneous, disease dynamics can be predicted by host habitat selection. Other diseases transmitted via environmental reservoirs such as avian influenza, chronic wasting disease and white-nose syndrome can also have spatial heterogeneity in disease risk as well as transmission hotspots on a landscape [61–63]. For these cases, because disease dynamics can be predicted by habitat use, movement of the host population can be a key component to understanding infection dynamics.

Ultimately, a long-term study would be required to monitor the host population growth rate as well as disease dynamics, to determine if this transmission hotspot is also an ecological trap. In addition, further study is needed to evaluate other potential mechanisms reinforcing a transmission hotspot, such as changes in host foraging behaviour [29,40]. Nevertheless, by investigating changes in habitat quality and animal movements, our study suggests that disease outbreaks can probably be determined by habitat dynamics and host resource selection when there is spatial heterogeneity in exposure to pathogens. Owing to shifts in host habitat selection in response to resource dynamics, the detrimental effect of the overlap in high-quality and high-risk

habitat on the host population is only intermittent. Thus, a heterogeneous landscape and environmental fluctuations may reduce the impact of an environmentally transmitted disease on a host population. With the fortuitous circumstances of the contrast of rainfall between the two study periods, our study contributes to better understanding disease dynamics in a natural system.

**Ethics.** Zebra capture protocols were approved by the Institutional Animal Care and Use Committee from the University of California, Berkeley (R217-0509B) and the University at Albany (16-016, 20-001).

**Data accessibility.** Fraction of photosynthetically active radiation (FPAR) data: moderate Resolution Imaging Spectroradiometer (Terra MODIS; MOD15A2H). R script and data (except for elephant and rhino mortality owing to sensitivity) of this study are available from the Dyrad Digital Repository: https://doi.org/10.5061/dryad. b2rbnzsdv [64]. Supporting information is provided in the electronic supplementary material.

**Authors' contributions.** Y.-H.H.: conceptualization, data curation, formal analysis, investigation, methodology, writing—original draft and writing—review and editing; H.J.: investigation and writing—review and editing; M.K.: investigation and writing—review and editing; Z.R.B.: investigation and writing—review and editing; C.C.C.: investigation and writing—review and editing; A.H.: investigation and writing—review and editing; P.L.K.: funding acquisition, project administration and writing—review and editing; W.K.: project administration, resources and writing—review and editing; J.K.E.M.: project administration and writing—review and editing; G.S.: investigation and writing—review and editing; R.Z.: investigation and writing—review and editing; W.M.G.: methodology, resources and writing—review and editing; W.C.T.: conceptualization, funding acquisition, investigation, methodology, project administration, supervision, writing—original draft and writing—review and editing. All authors gave final approval for publication and agreed to be held accountable for the work performed therein.

**Competing interests.** We declare we have no competing interests.

**Funding.** This work was supported by NSF Division of Environment Biology (DEB-1816161/DEB-2106221) to W.C.T., P.L.K. and Henriette van Heerden, and funds from the University at Albany. The data collection from 2009 to 2010 was supported by NIH National Institute of General Medical Sciences (GM083863) to W.M.G. P.L.K was supported by the USDA National Institute of Food and Agriculture Hatch project (no. ME021908), through the Maine Agricultural and Forest Experiment Station. Any use of trade, firm or product names is for descriptive purposes only and does not imply endorsement by the US Government.

**Acknowledgements.** We thank the Namibian National Commission on Research, Science and Technology for permission to conduct this research (authorization 2017070704). We especially thank the Ministry of Environment, Forestry and Tourism and the Etosha Ecological Institute for support of and assistance with our research activities. Seth Guim, Jason Iiyambo, Naftali Iiyambo, Mark Jago, Johannes Kapner, Kantana Mathews, Carl-Heinz Moeller, Paulus Namholo and Janine Sharpe provided assistance with animal captures and data collection. We also acknowledge helpful comments from Paul Cross, Norman Owen-Smith and an anonymous reviewer.

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
