## [Peer Review File · Proceedings of the Royal Society B: Biological Sciences]

Review History

RSPB-2020-2709.R0 (Original submission)

Review form: Reviewer 1 (Norman Owen-Smith)

Recommendation

Accept with minor revision (please list in comments)

Scientific importance: Is the manuscript an original and important contribution to its field?

Good

General interest: Is the paper of sufficient general interest?

Good

Quality of the paper: Is the overall quality of the paper suitable?

Excellent

Is the length of the paper justified?

Yes

Should the paper be seen by a specialist statistical reviewer?

No

Do you have any concerns about statistical analyses in this paper? If so, please specify them explicitly in your report.

No

It is a condition of publication that authors make their supporting data, code and materials available - either as supplementary material or hosted in an external repository. Please rate, if applicable, the supporting data on the following criteria.

Is it accessible?

N/A

Is it clear?

N/A

Is it adequate?

N/A

Do you have any ethical concerns with this paper?

No

Comments to the Author

The authors observed a large reduction in mortality of zebra caused by anthrax between years of average rainfall and a drought year when rainfall was greatly reduced. They related this to a shift in habitat selection by zebra during the drought from open plains to woodland during the wet season when most anthrax deaths occur. The implication drawn was that a population can be intermittently released from the 'ecological trap' that may be formed when habitats favourable nutritionally are also most infectious. Overall, the findings emphasise the importance of spatial heterogeneity for facilitating population persistence in environments experiencing wide temporal variation, a principle not sufficiently widely recognised.

The main limitation of the study is the 10-year gap in time from when the initial observations were made during an average year and the later investigation under exceptionally extreme drought conditions. It is possible that the infectiousness of anthrax had faded in the interim for reasons unrelated to habitat selection by the zebra. Deaths due to anthrax were exceptionally high during the wet season of the initial study. Although this situation potentially exaggerates the contrast between the two periods, the difference in mortalities remains huge. In the final year, deaths due to anthrax did not approach their former level reached a decade earlier, despite above-average rainfall. It is unfortunate that a comparison could not be drawn between the mortality pattern shown by zebra and those experienced by other ungulate species not showing a habitat switch. This limitation is due to the overwhelming concentration of deaths due to anthrax on zebra, for reasons explained in other publications. Nevertheless, the authors were fortunate in having follow-up observations undertaken during the period spanning the drought extreme. The study makes an important contribution to understanding of disease ecology due to these fortuitous circumstances.

Not adequately explained is why deaths due to anthrax did not increase to their former levels in the wet season of 2020 when the zebra concentrated as strongly in open habitat as they had done a decade earlier. Anthrax spores persist for many years in the soil and so should not fade substantially during one very dry year unless other processes were operating. What other factors such as 'population density' or 'animal behaviour' could explain this contrast? Hysteresis effects can be very important for ecosystem dynamics, and mechanisms potentially generating them for disease dynamics need to be postulated for further investigation as time plays out.

Bigger puzzles in anthrax ecology still remain unresolved. Why do zebra incur virtually no mortality due to anthrax in Kruger Park, despite similar habitats to Etosha, while kudus that hardly feature in Etosha are among the main sufferers in Kruger?

Norman Owen-Smith

Review form: Reviewer 2

Recommendation

Major revision is needed (please make suggestions in comments)

Scientific importance: Is the manuscript an original and important contribution to its field?

Good

General interest: Is the paper of sufficient general interest?

Excellent

Quality of the paper: Is the overall quality of the paper suitable?

Good

Is the length of the paper justified?

Yes

Should the paper be seen by a specialist statistical reviewer?

No

Do you have any concerns about statistical analyses in this paper? If so, please specify them explicitly in your report.

No

It is a condition of publication that authors make their supporting data, code and materials available - either as supplementary material or hosted in an external repository. Please rate, if applicable, the supporting data on the following criteria.

Is it accessible?

Yes

Is it clear?

Yes

Is it adequate?

Yes

Do you have any ethical concerns with this paper?

No

Comments to the Author

This study presents an empirical example of the ecological trap concept, using a system in which high quality habitat appears to be associated with high pathogen infection risk. The authors argue that spatio-temporal variation in habitat quality due to fluctuations in environmental conditions results in hosts selecting alternative habitats during other times and to thereby achieve escape from high disease risk i.e. the ecological trap.

I found the study interesting and well presented. However, I didn't become fully convinced that the data presented are really sufficient to support the authors' conclusions. There are several points to this:

First, the authors argue that spending time in open habitats is associated with a higher infection risk compared to closed habitats. However, as they acknowledge, carcass detection probability is likely to vary among habitat types, which makes it difficult to know whether higher numbers of carcasses actually reflect higher mortality. Confusingly, the authors mention in the Methods that they 'estimated spatially relative anthrax risk by habitat type by comparing anthrax mortalities to mortalities from other natural causes' - however, as far as I can see the results only show absolute numbers, not relative mortality risk. In addition to detection probability, other factors might contribute to high number of anthrax carcasses e.g. changes in population size (which is acknowledged in the Discussion). So a stronger case needs to be made that the higher anthrax mortality risk in open habitats is real and not just an artefact.

Secondly, the manuscript repeatedly makes reference to population level effects despite the fact that no population data are presented (e.g. L72 - 'Periods of environmental fluctuation that allow reprieve from the transmission hotspot will support population persistence' - we have no idea whether the results presented here have any relevance to population persistence). I would suggest to remove these references or at least to make clear that these extended consequences are poorly speculative. Even at the individual level, we ultimately don't know about the relative costs (infection risk) and benefits (i.e. fitness gains) from feeding in high quality habitat to create the hypothesised trade-off. Any data that could be added to further illustrate or even quantify this tradeoff would strengthen the paper.

The authors seem to largely be aware of these limitations, given that they are brought up at some length towards the end of the Discussion. I would suggest to rewrite the manuscript with these caveats in mind.

Additional specific comments:

1) As I understand it, the authors did not include information on where carcasses were found. Instead their argument is based on higher numbers of mortality events during periods when open habitats were preferred. This seems fine given the incubation period of the pathogen (several days?), which means the animal could have moved over a significant distance between where it was exposed and where it died. The authors speculate (L247) that 'zebras contracting the disease on the plains will probably die in these areas and create more local infectious zones'. Could the detailed movement data they collected be used to substantiate this? For example, by looking at consistency in habitat selection over the length of the average time period between exposure and death?

2) L61 - 'Given that anthrax is highly lethal, the fitness consequence for an animal population is more substantial'

Not a great sentence. Fitness as a concept applies to individuals rather than populations and the fact that anthrax is commonly lethal, on its own, does not tell us anything about how it affects the host population.

3) L98 'Areas with no vegetation type assigned were removed from this study.'
What percentage of data points did this apply to?

4) more information needed about how cause of mortality was determined - is the probability of being designated mortality due to anthrax really independent of habitat?

5) L144 'thinned the data by extracting the reading closest to 9:00, 15:00 and 24:00 for the three periods of a day'.

How variable was habitat selection within each time period - were results robust to this thinning

approach or was there a lot of noise?

6) L203 'zebra predominantly used open plains over any closed habitats in the wet season of years with average and above average rainfall when there was higher vegetation productivity (figures 4a and 4b).

This information cannot be gleaned from Fig 4, which does not include data on rainfall. I would suggest to visually indicate which observations in Figs 4 & 5 are from above average rainfall periods. But the statement above is also not backed up by any statistical evidence - how does this relate to results from the regression models?

7) Fig 3 doesn't seem very color-blind friendly - choose different colour scheme?

Decision letter (RSPB-2020-2709.R0)

21-Jan-2021

Dear Mr Huang:

I am writing to inform you that your manuscript RSPB-2020-2709 entitled "Disease or drought: environmental fluctuations release zebra from a potential pathogen-triggered ecological trap" has, in its current form, been rejected for publication in Proceedings B.

This action has been taken on the advice of referees, who have recommended that substantial revisions are necessary. With this in mind we would be happy to consider a resubmission, provided the comments of the referees are fully addressed. However please note that this is not a provisional acceptance.

Sincerely,
 Professor Hans Heesterbeek
 mailto: proceedingsb@royalsociety.org

Associate Editor

Board Member: 1

Comments to Author:

This paper presents results of a largely correlative study on zebra in Namibia that examines how habitat preference for resources interacts with risk due to anthrax. It is couched in terms of an 'ecological trap', with the idea being that preference for resources in a high-risk area can cause population decline. The authors find that habitat preference varies temporally due to external conditions and so this can provide a reprieve for individuals from areas where infection risk is high.

I think this study is interesting and both referees did as well. However, I think there are a few comments that need to be addressed before I would recommend publication. From my reading these include being more cautious about the interpretation that mortality risk does really vary as inferred (because of limitations in the data, carcass detection, etc) and also being more cautious that the observed patterns really do translate into population-level consequences. Both of these issues are discussed already in the paper but highlighting them a bit more when the methodology and results are being presented is probably warranted. Finally, I think it is worth saying a bit more about the concern that one referees had about other confounding effects playing a role in the results given the large temporal period in question.

Reviewer(s)' Comments to Author:

Referee: 1

Comments to the Author(s)

The authors observed a large reduction in mortality of zebra caused by anthrax between years of average rainfall and a drought year when rainfall was greatly reduced. They related this to a shift in habitat selection by zebra during the drought from open plains to woodland during the wet season when most anthrax deaths occur. The implication drawn was that a population can be intermittently released from the 'ecological trap' that may be formed when habitats favourable nutritionally are also most infectious. Overall, the findings emphasise the importance of spatial heterogeneity for facilitating population persistence in environments experiencing wide temporal variation, a principle not sufficiently widely recognised.

The main limitation of the study is the 10-year gap in time from when the initial observations were made during an average year and the later investigation under exceptionally extreme drought conditions. It is possible that the infectiousness of anthrax had faded in the interim for reasons unrelated to habitat selection by the zebra. Deaths due to anthrax were exceptionally high during the wet season of the initial study. Although this situation potentially exaggerates the contrast between the two periods, the difference in mortalities remains huge. In the final year, deaths due to anthrax did not approach their former level reached a decade earlier, despite above-average rainfall. It is unfortunate that a comparison could not be drawn between the mortality pattern shown by zebra and those experienced by other ungulate species not showing a habitat switch. This limitation is due to the overwhelming concentration of deaths due to anthrax on zebra, for reasons explained in other publications. Nevertheless, the authors were fortunate in having follow-up observations undertaken during the period spanning the drought extreme. The study makes an important contribution to understanding of disease ecology due to these fortuitous circumstances.

Not adequately explained is why deaths due to anthrax did not increase to their former levels in the wet season of 2020 when the zebra concentrated as strongly in open habitat as they had done a decade earlier. Anthrax spores persist for many years in the soil and so should not fade substantially during one very dry year unless other processes were operating. What other factors such as 'population density' or 'animal behaviour' could explain this contrast? Hysteresis effects can be very important for ecosystem dynamics, and mechanisms potentially generating them for disease dynamics need to be postulated for further investigation as time plays out.

Bigger puzzles in anthrax ecology still remain unresolved. Why do zebra incur virtually no mortality due to anthrax in Kruger Park, despite similar habitats to Etosha, while kudus that hardly feature in Etosha are among the main sufferers in Kruger?

Norman Owen-Smith

Referee: 2

Comments to the Author(s)

This study presents an empirical example of the ecological trap concept, using a system in which high quality habitat appears to be associated with high pathogen infection risk. The authors argue that spatio-temporal variation in habitat quality due to fluctuations in environmental conditions results in hosts selecting alternative habitats during other times and to thereby achieve escape from high disease risk i.e. the ecological trap.

I found the study interesting and well presented. However, I didn't become fully convinced that the data presented are really sufficient to support the authors' conclusions. There are several points to this:

First, the authors argue that spending time in open habitats is associated with a higher infection risk compared to closed habitats. However, as they acknowledge, carcass detection probability is likely to vary among habitat types, which makes it difficult to know whether higher numbers of carcasses actually reflect higher mortality. Confusingly, the authors mention in the Methods that they 'estimated spatially relative anthrax risk by habitat type by comparing anthrax mortalities to mortalities from other natural causes' - however, as far as I can see the results only show absolute numbers, not relative mortality risk. In addition to detection probability, other factors might contribute to high number of anthrax carcasses e.g. changes in population size (which is acknowledged in the Discussion). So a stronger case needs to be made that the higher anthrax mortality risk in open habitats is real and not just an artefact.

Secondly, the manuscript repeatedly makes reference to population level effects despite the fact that no population data are presented (e.g. L72 - 'Periods of environmental fluctuation that allow reprieve from the transmission hotspot will support population persistence' - we have no idea whether the results presented here have any relevance to population persistence). I would suggest to remove these references or at least to make clear that these extended consequences are poorly speculative. Even at the individual level, we ultimately don't know about the relative costs (infection risk) and benefits (i.e. fitness gains) from feeding in high quality habitat to create the hypothesised trade-off. Any data that could be added to further illustrate or even quantify this tradeoff would strengthen the paper.

The authors seem to largely be aware of these limitations, given that they are brought up at some length towards the end of the Discussion. I would suggest to rewrite the manuscript with these caveats in mind.

Additional specific comments:

1) As I understand it, the authors did not include information on where carcasses were found. Instead their argument is based on higher numbers of mortality events during periods when open habitats were preferred. This seems fine given the incubation period of the pathogen (several days?), which means the animal could have moved over a significant distance between where it was exposed and where it died. The authors speculate (L247) that 'zebras contracting the disease on the plains will probably die in these areas and create more local infectious zones'. Could the detailed movement data they collected be used to substantiate this? For example, by looking at consistency in habitat selection over the length of the average time period between exposure and death?

- 2) L61 - 'Given that anthrax is highly lethal, the fitness consequence for an animal population is more substantial'
Not a great sentence. Fitness as a concept applies to individuals rather than populations and the fact that anthrax is commonly lethal, on its own, does not tell us anything about how it affects the host population.
- 3) L98 'Areas with no vegetation type assigned were removed from this study.'
What percentage of data points did this apply to?
- 4) more information needed about how cause of mortality was determined - is the probability of being designated mortality due to anthrax really independent of habitat?
- 5) L144 'thinned the data by extracting the reading closest to 9:00, 15:00 and 24:00 for the three periods of a day'.
How variable was habitat selection within each time period - were results robust to this thinning approach or was there a lot of noise?
- 6) L203 'zebra predominantly used open plains over any closed habitats in the wet season of years with average and above average rainfall when there was higher vegetation productivity (figures 4a and 4b).
This information cannot be gleaned from Fig 4, which does not include data on rainfall. I would suggest to visually indicate which observations in Figs 4 & 5 are from above average rainfall periods. But the statement above is also not backed up by any statistical evidence - how does this relate to results from the regression models?
- 7) Fig 3 doesn't seem very color-blind friendly - choose different colour scheme?

Author's Response to Decision Letter for (RSPB-2020-2709.R0)

See Appendix A.

RSPB-2021-0582.R1 (Revision)

Review form: Reviewer 1 (Norman Owen-Smith)

Recommendation

Accept as is

Scientific importance: Is the manuscript an original and important contribution to its field?

Good

General interest: Is the paper of sufficient general interest?

Good

Quality of the paper: Is the overall quality of the paper suitable?

Excellent

Is the length of the paper justified?

Yes

Should the paper be seen by a specialist statistical reviewer?

No

Do you have any concerns about statistical analyses in this paper? If so, please specify them explicitly in your report.

No

It is a condition of publication that authors make their supporting data, code and materials available - either as supplementary material or hosted in an external repository. Please rate, if applicable, the supporting data on the following criteria.

Is it accessible?

Yes

Is it clear?

Yes

Is it adequate?

Yes

Do you have any ethical concerns with this paper?

No

Comments to the Author

I have no further comments to make

Decision letter (RSPB-2021-0582.R0)

05-May-2021

Dear Mr Huang

I am pleased to inform you that your Review manuscript RSPB-2021-0582 entitled "Disease or drought: environmental fluctuations release zebra from a potential pathogen-triggered ecological trap" has been accepted for publication in Proceedings B.

The referee and the Associate Editor do not recommend any further changes. Therefore, please proof-read your manuscript carefully and upload your final files for publication. Because the schedule for publication is very tight, it is a condition of publication that you submit the revised version of your manuscript within 7 days. If you do not think you will be able to meet this date please let me know immediately.

To upload your manuscript, log into <http://mc.manuscriptcentral.com/prsb> and enter your Author Centre, where you will find your manuscript title listed under "Manuscripts with Decisions." Under "Actions," click on "Create a Revision." Your manuscript number has been appended to denote a revision.

You will be unable to make your revisions on the originally submitted version of the manuscript. Instead, upload a new version through your Author Centre.

1) A text file of the manuscript (doc, txt, rtf or tex), including the references, tables (including captions) and figure captions. Please remove any tracked changes from the text before submission. PDF files are not an accepted format for the "Main Document".

2) A separate electronic file of each figure (tiff, EPS or print-quality PDF preferred). The format should be produced directly from original creation package, or original software format. Please note that PowerPoint files are not accepted.

3) Electronic supplementary material: this should be contained in a separate file from the main text and the file name should contain the author's name and journal name, e.g. `authorname_procb_ESM_figures.pdf`

All supplementary materials accompanying an accepted article will be treated as in their final form. They will be published alongside the paper on the journal website and posted on the online figshare repository. Files on figshare will be made available approximately one week before the accompanying article so that the supplementary material can be attributed a unique DOI. Please see: <https://royalsociety.org/journals/authors/author-guidelines/>

4) Data-Sharing and data citation

It is a condition of publication that data supporting your paper are made available. Data should be made available either in the electronic supplementary material or through an appropriate repository. Details of how to access data should be included in your paper. Please see <https://royalsociety.org/journals/ethics-policies/data-sharing-mining/> for more details.

<http://datadryad.org/submit?journalID=RSPB&manu=RSPB-2021-0582> which will take you to your unique entry in the Dryad repository.

Once again, thank you for submitting your manuscript to Proceedings B and I look forward to receiving your final version. If you have any questions at all, please do not hesitate to get in touch.

Sincerely,

Professor Hans Heesterbeek

Reviewer(s)' Comments to Author:

Referee: 1

Comments to the Author(s).

I have no further comments to make

Decision letter (RSPB-2021-0582.R1)

10-May-2021

Dear Mr Huang

I am pleased to inform you that your manuscript entitled "Disease or drought: environmental fluctuations release zebra from a potential pathogen-triggered ecological trap" has been accepted for publication in Proceedings B.

If you are likely to be away from e-mail contact please let us know. Due to rapid publication and an extremely tight schedule, if comments are not received, we may publish the paper as it stands. If you have any queries regarding the production of your final article or the publication date please contact procb_proofs@royalsociety.org

Data Accessibility section

Open Access

Paper charges

Sincerely,

Proceedings B

Appendix A

Response to referees

Reviewer comments in blue, our response in black.

Associate Editor

Board Member: 1

Comments to Author:

This paper presents results of a largely correlative study on zebra in Namibia that examines how habitat preference for resources interacts with risk due to anthrax. It is couched in terms of an ‘ecological trap’, with the idea being that preference for resources in a high-risk area can cause population decline. The authors find that habitat preference varies temporally due to external conditions and so this can provide a reprieve for individuals from areas where infection risk is high.

I think this study is interesting and both referees did as well. However, I think there are a few comments that need to be addressed before I would recommend publication. From my reading these include being more cautious about the interpretation that mortality risk does really vary as inferred (because of limitations in the data, carcass detection, etc) and also being more cautious that the observed patterns really do translate into population-level consequences. Both of these issues are discussed already in the paper but highlighting them a bit more when the methodology and results are being presented is probably warranted. Finally, I think it is worth saying a bit more about the concern that one referees had about other confounding effects playing a role in the results given the large temporal period in question.

In response to the editor’s recommendations and comments from the referees on being cautious and mentioning caveats earlier, we clarified that the focus of this study is on how host habitat selection varies with resource availability, which in turn affects disease risk and outbreak dynamics both seasonally and inter-annually (L39-41; L262-269). We still frame this in the context of the ecological trap literature, but clarify in the introduction that we do not have the fitness data required to test for an ecological trap (L46-48). We further added additional data and analyses in the supplementary materials to clarify and support our assessment of differential disease risk among habitats, and why the temporal gap between the two study periods should not confound the results of the study (more detail is provided below in response to specific review comments). We believe that these changes have strengthened the paper, and we appreciate the efforts of the editor and the two reviewers.

Reviewer(s)' Comments to Author:

Referee: 1

Comments to the Author(s)

The authors observed a large reduction in mortality of zebra caused by anthrax between years of average rainfall and a drought year when rainfall was greatly reduced. They related this to a shift in habitat selection by zebra during the drought from open plains to woodland during the wet season when most anthrax deaths occur. The implication drawn was that a population can be intermittently released from the 'ecological trap' that may be formed when habitats favourable nutritionally are also most infectious. Overall, the findings emphasise the importance of spatial heterogeneity for facilitating population persistence in environments experiencing wide temporal variation, a principle not sufficiently widely recognised.

The main limitation of the study is the 10-year gap in time from when the initial observations were made during an average year and the later investigation under exceptionally extreme drought conditions. It is possible that the infectiousness of anthrax had faded in the interim for reasons unrelated to habitat selection by the zebra. Deaths due to anthrax were exceptionally high during the wet season of the initial study. Although this situation potentially exaggerates the contrast between the two periods, the difference in mortalities remains huge. In the final year, deaths due to anthrax did not approach their former level reached a decade earlier, despite above-average rainfall. It is unfortunate that a comparison could not be drawn between the mortality pattern shown by zebra and those experienced by other ungulate species not showing a habitat switch. This limitation is due to the overwhelming concentration of deaths due to anthrax on zebra, for reasons explained in other publications. Nevertheless, the authors were fortunate in having follow-up observations undertaken during the period spanning the drought extreme. The study makes an important contribution to understanding of disease ecology due to these fortuitous circumstances.

Anthrax cases have been recorded every year in the study system since 1964 when the disease was first identified. Thus, new infectious sites are added to the system each year, and the 10-year gap does not mean that the infectious sites created in the first study period need to carry over to the second study period for new cases to occur. To clarify this, we added the following text: "Zebra anthrax mortalities occur annually with strong seasonality in cases, which peak in the late wet season (March – April) [29, 31], and case numbers are positively correlated with annual rainfall [29] (figure S1b)."

(L101-103)

That said, there is a positive correlation between annual rainfall and the number of zebra anthrax cases recorded (Turner et al. 2013 Ecosphere). Thus, multi-year periods of above average rainfall or below average rainfall will translate into periods of more or less zebra anthrax in the system, and that may in part explain why the number of anthrax cases after the drought did not reach the levels seen in 2010 (more on this in response to next comment below).

The comment about using several host species to make inferences about the dynamics of this multi-host wildlife disease is an important point. We were unable to do that in this particular study, since we lacked the matched data. However, the relationships between rainfall, resource availability and anthrax mortality patterns for four host species is the focus of a follow up study currently in preparation.

Not adequately explained is why deaths due to anthrax did not increase to their former levels in the wet season of 2020 when the zebra concentrated as strongly in open habitat as they had done a decade earlier. Anthrax spores persist for many years in the soil and so should not fade substantially during one very dry year unless other processes were operating. What other factors such as ‘population density’ or ‘animal behaviour’ could explain this contrast? Hysteresis effects can be very important for ecosystem dynamics, and mechanisms potentially generating them for disease dynamics need to be postulated for further investigation as time plays out.

This a great point. We added a supplementary figure (S1) of rainfall and anthrax mortality patterns over time, and additional text (L312-319) suggesting that the variation in the size of the outbreaks may be due to multi-year rainfall patterns:

“The variation in outbreak sizes between 2010 and 2020 may be attributed to factors such as multi-year rainfall patterns. Though spores can persist in the environment for decades, spore concentrations at a site decay over time, and carcass sites are most infectious within the first few years after host death [21]. Zebra anthrax cases are positively correlated with annual rainfall [29]. Hence, periods of above average rainfall and higher zebra anthrax mortality will increase the number of highly infectious reservoir sites, while dry periods will deplete them. High rainfall years with more zebra anthrax cases preceded the large outbreak recorded in 2010, while drier years with fewer anthrax cases preceded the smaller outbreak in 2020 (figure S1a).”

Bigger puzzles in anthrax ecology still remain unresolved. Why do zebra incur virtually no mortality due to anthrax in Kruger Park, despite similar habitats to Etosha, while kudus that hardly feature in Etosha are among the main sufferers in Kruger?

These are excellent questions, and are at the heart of what drive our larger research program comparing anthrax dynamics in Etosha and Kruger National Parks. Our current hypotheses run the gamut of these systems having different transmission pathways, different host densities, different home range sizes (affecting spatial spread when outbreaks occur), different seasonality of exposure, and different suitability of the environment for spore persistence. We hope, within the next few years, to have answers!

Norman Owen-Smith

Referee: 2

Comments to the Author(s)

This study presents an empirical example of the ecological trap concept, using a system in which high quality habitat appears to be associated with high pathogen infection risk. The authors argue that spatio-temporal variation in habitat quality due to fluctuations in environmental conditions results in hosts selecting alternative habitats during other times and to thereby achieve escape from high disease risk i.e. the ecological trap.

I found the study interesting and well presented. However, I didn't become fully convinced that the data presented are really sufficient to support the authors' conclusions. There are several points to this:

First, the authors argue that spending time in open habitats is associated with a higher infection risk compared to closed habitats. However, as they acknowledge, carcass detection probability is likely to vary among habitat types, which makes it difficult to know whether higher numbers of carcasses actually reflect higher mortality. Confusingly, the authors mention in the Methods that they 'estimated spatially relative anthrax risk by habitat type by comparing anthrax mortalities to mortalities from other natural causes' - however, as far as I can see the results only show absolute numbers, not relative mortality risk. In addition to detection probability, other factors might contribute to high number of anthrax carcasses e.g. changes in population size (which is acknowledged in the Discussion). So a

stronger case needs to be made that the higher anthrax mortality risk in open habitats is real and not just an artefact.

We were not clear in how we presented the risk analysis, and addressed this comment by clarifying our language in the methods (L107-122) and results (L226-228), by adding the ratios of anthrax to other causes of death to figure 2, and by adding figure S2 which corroborates the relative risk patterns detected in the larger, opportunistic surveillance dataset using mortality records collected during periods of enhanced surveillance effort. These numbers/analyses demonstrate that open plains had higher ratios of anthrax mortalities to other mortalities than closed habitats, and that woodlands had the lowest ratio (figures 2 and S2).

Secondly, the manuscript repeatedly makes reference to population level effects despite the fact that no population data are presented (e.g. L72 - 'Periods of environmental fluctuation that allow reprieve from the transmission hotspot will support population persistence' - we have no idea whether the results presented here have any relevance to population persistence). I would suggest to remove these references or at least to make clear that these extended consequences are poorly speculative. Even at the individual level, we ultimately don't know about the relative costs (infection risk) and benefits (i.e. fitness gains) from feeding in high quality habitat to create the hypothesised trade-off. Any data that could be added to further illustrate or even quantify this tradeoff would strengthen the paper.

This is a good point. We clarified in the introduction that we are framing this study in the context of ecological traps, but that our study is not specifically testing for an ecological trap:

“Although an earlier demographic study concluded that anthrax may be limiting growth of this zebra population [19], we cannot assess whether an anthrax transmission hotspot is also an ecological trap, because we did not measure fitness differences among habitats.” (L46-48)

As for the quoted sentence previously at L72 (and elsewhere), we adjusted our language to remove the speculation about population persistence. We agree that without evidence, the population effects of a trade-off between costs and benefits are unknown, and that support of population persistence is purely speculative. We changed inferences on population persistence to a reprieve from the transmission hotspot:

“Periods of environmental fluctuation that allow reprieve from the transmission hotspot will reduce exposure and the number of pathogen-induced mortalities affecting the population.” (L62-64)

“Thus, a heterogeneous landscape and environmental fluctuations may reduce the impact of an environmentally transmitted disease on a host population.” (L336-338)

The authors seem to largely be aware of these limitations, given that they are brought up at some length towards the end of the Discussion. I would suggest to rewrite the manuscript with these caveats in mind.

Given this comment and the editor’s suggestions, we added additional text to the methods and additional results/analyses to clarify how we assessed relative risk among habitats. We further clarified in the introduction and discussion that ecological traps are not tested in this study due to the lack of fitness data (L39-41; L46-48; L279-280).

Additional specific comments:

1) As I understand it, the authors did not include information on where carcasses were found. Instead their argument is based on higher numbers of mortality events during periods when open habitats were preferred. This seems fine given the incubation period of the pathogen (several days?), which means the animal could have moved over a significant distance between where it was exposed and where it died. The authors speculate (L247) that 'zebras contracting the disease on the plains will probably die in these areas and create more local infectious zones'. Could the detailed movement data they collected be used to substantiate this? For example, by looking at consistency in habitat selection over the length of the average time period between exposure and death?

The GPS location information for carcasses (anthrax or otherwise) was used to assess relative risk by habitat type (data shown in figure 2) and for zebra anthrax cases by habitat type and season (for figure 3). We did not try to link carcass locations to movement tracks of monitored animals. The very different spatial and temporal scales for these datasets makes this a particularly challenging task requiring methodological development at the intersection of movement and disease ecology, and is beyond the scope of this study. How GPS location information for carcasses was used is stated in the methods (L123-133; L135-139).

As for the second point, this is a great idea. We added an analysis in supplementary materials determining the probability over time that zebra individuals present in the open habitat remained in the open habitat. We evaluated this for a 10-day period, covering the expected incubation period

between exposure and death (estimated to occur as quickly as a few days to a week). The probability that animals remained in open habitats was highest in the wet/anthrax season (figure S5).

2) L61 - 'Given that anthrax is highly lethal, the fitness consequence for an animal population is more substantial'

Not a great sentence. Fitness as a concept applies to individuals rather than populations and the fact that anthrax is commonly lethal, on its own, does not tell us anything about how it affects the host population.

Population-level fitness is an integration across individual fitnesses (Edelaar and Bolnick 2019 Trends in Ecology & Evolution). To better clarify the idea of mean fitness (across individuals) in a population, we edited this sentence to “Given its lethality, anthrax has potential fitness consequences for the zebra population, by reducing average fitness of individuals.” (L44-46). There is evidence from a previous study, cited in this paragraph, that zebra population growth may be limited by anthrax in this system.

3) L98 'Areas with no vegetation type assigned were removed from this study.'

What percentage of data points did this apply to?

The percentage of removed thinned readings due to no vegetation assigned was 6.8. We added this in the methods (L162-163).

4) more information needed about how cause of mortality was determined - is the probability of being designated mortality due to anthrax really independent of habitat?

A diagnosis of anthrax would not be affected by the habitat in which the carcass was found. The laboratory diagnostics for anthrax confirmation are independent of observer and independent of suspected cause of death. Mortalities are tested for anthrax whether or not the animal is a suspected anthrax mortality. We added more details how an anthrax diagnosis was determined to clarify this in the methods (L95-103).

5) L144 'thinned the data by extracting the reading closest to 9:00, 15:00 and 24:00 for the three periods of a day'.

How variable was habitat selection within each time period - were results robust to this thinning approach or was there a lot of noise?

We added an analysis in supplementary materials comparing the results using the thinning methods applied in this study and for fixed intervals ranging from 1 to 24 hours. The results show that the thinning approach we used was consistent with fixed intervals between 1 to 8 hours, except for bare area usage for some individuals, probably due to infrequent use in the area (figure S4). We made note of this in the methods (L180-183).

6) L203 'zebra predominantly used open plains over any closed habitats in the wet season of years with average and above average rainfall when there was higher vegetation productivity (figures 4a and 4b).

This information cannot be gleaned from Fig 4, which does not include data on rainfall. I would suggest to visually indicate which observations in Figs 4 & 5 are from above average rainfall periods. But the statement above is also not backed up by any statistical evidence - how does this relate to results from the regression models?

We added color-coded FPAR levels by season to boxplots in figure 4 to better visualize environmental fluctuations. We also added colour-/shape-coded years/seasons in figures 5 and 6 to better identify the rainfall years. Figure 5 now provides clearer information that zebra selected open habitats more dominantly in the wet seasons of 2010 and 2020. We also mentioned figure 5 when describing zebra seasonal preferences in habitat selection in the results (L246-251).

7) Fig 3 doesn't seem very color-blind friendly - choose different colour scheme?

We changed the colours in figure 1 and 3 to be more colour-blind friendly.